# Modeling the Effects of Temperature and Limiting Nutrients on the Competition of an Invasive Floating Plant, *Pontederia crassipes*, with Submersed Vegetation in a Shallow Lake

**DOI:** 10.3390/plants13182621

**Published:** 2024-09-20

**Authors:** Linhao Xu, Donald L. DeAngelis

**Affiliations:** 1Department of Biology, University of Miami, Coral Gables, FL 33124, USA; linhao.xu@miami.edu; 2Wetland and Aquatic Research Center, U. S., Geological Survey, Davie, FL 33314, USA

**Keywords:** spatially explicit model, water hyacinth, Nuttall’s waterweed, temperature seasonality, nutrient limitation, bifurcation diagram

## Abstract

The potential for a non-native plant species to invade a new habitat depends on broadscale factors such as climate, local factors such as nutrient availability, and the biotic community of the habitat into which the plant species is introduced. We developed a spatially explicit model to assess the risk of expansion of a floating invasive aquatic plant species (FAV), the water hyacinth (*Pontederia crassipes*), an invader in the United States, beyond its present range. Our model used known data on growth rates and competition with a native submersed aquatic macrophyte (SAV). In particular, the model simulated an invasion into a habitat with a mean annual temperature different from its own growth optimum, in which we also simulated seasonal fluctuations in temperature. Twenty different nutrient concentrations and eight different temperature scenarios, with different mean annual amplitudes of seasonal temperature variation around the mean of the invaded habitat, were simulated. In each case, the ability of the water hyacinth to invade and either exclude or coexist with the native vegetation was determined. As the temperature pattern was changed from tropical towards increasingly cooler temperate levels, the competitive advantage shifted from the tropical FAV to the more temperate SAV, with a wide range in which coexistence occurred. High nutrient concentrations allowed the coexistence of FAV, even at cooler annual temperatures. But even at the highest nutrient concentrations in the model, the FAV was unlikely to persist under the current climates of latitudes in the Southeastern United States above that of Northern Alabama. This result may have some implications for where control efforts need to be concentrated.

## 1. Introduction

Invasive plant species threaten native vegetation and agricultural systems around the world [1] and can cause significant economic harm to native biodiversity [2,3,4]. Success in a new environment includes the ability to escape from specialist natural enemies [5,6], wherein invasive species may grow and reproduce in an unchecked fashion. It is also believed that native species are weaker competitors than invaders in a niche, though this has not been proven [7,8]. This can lead to regime shifts that can be hard to reverse due to hysteresis effects, even with control efforts [9]. The problems posed by invasive species will be exacerbated by climate change [10,11].

Water hyacinth ((*Pontederia crassipes*) (Mart.) Solms (Pontederiaceae), formerly *Eichhornia crassipes*) native to the Amazon Basin of South America [12,13,14], is considered invasive in over 50 countries in North America, Africa, Europe, Asia, and Australia [15,16] and has been called one of the ten worst weeds in the world [17,18]. In Southern Florida, water hyacinth is a threat to several natural wetland habitats, including the Everglades. Water hyacinth prefers slow-moving water, full sun, warm temperature, and high nutrient concentrations, particularly regarding nitrogen. At maturity, it spreads via seeds and stolons [19]. Water hyacinth reduces oxygen exchange in waterbodies by forming thick mats over water surfaces [20], excluding native vegetation such as submersed macrophytes [21], and disrupting food webs [22]. For the manual removal of water hyacinth, chemical and biological control methods have been utilized, with varying success [23]. For biological control, the mottled water hyacinth weevil, *Neochetina eichhorniae* Warner (Coleoptera: Curculionidae), and the chevron water hyacinth weevil, *N. bruchii*, were released in Southern Florida in the late 1970s and managed to significantly reduce biomass and reproduction, lowering the fitness of water hyacinth overall in many habitats but only partly reducing surface coverage [24]. The development of effective sustainable control measures for water hyacinth is essential both economically and for environmental conservation. Large information gaps remain concerning the most effective means for control, so further empirical and modeling research is needed.

Due to the current spread of invasive species and their projected spread from the worldwide transport of biological matter and changing habitat conditions due to climate change, modeling approaches have been developed to estimate the potential spread of species beyond their native ranges [25,26,27,28,29]. The two major types of models in this regard are mechanistic and correlative. Both are related to the idea of a species’ ‘fundamental niche’, a hypervolume of all the resource requirements and tolerances to stresses of the species [30]. A species can survive and reproduce in geographic areas where sufficient resources and stress conditions are within that hypervolume, termed a ‘potential niche’ [31]. The ‘realized niche’ of geographic area in which a species can actually survive may be smaller due to other environmental factors such as competitors and predators [32].

Mechanistic modeling, in which the biophysical and behavioral characteristics of a species are taken into account, can describe the fundamental niche of a species [33] and is an active research area, but it can involve high data requirements and complex models. Therefore, there has been a heavy emphasis on correlative models, which have been given various names, such as species distribution models (SDMs), ecological niche models, and bioclimate envelopes [34]. For example, SDMs, using data from remote sensing, spatial analysis, and computer models such as MaxEnt (e.g., latest version 3.3.4), are widely used in conservation [35,36,37,38]. These models can map ecologically suitable habitats through estimating the relationship between species presence data and the drivers through statistical algorithms under comprehensive environmental conditions [39] and thus have been widely used to predict potentially suitable habitats for species. The conditions used to assess the probability of establishment usually include a number of coarse-scale indices mostly representing climatic conditions. But they can also include spatially finer-scale variables, such as soil type, elevation, slope, distance to rivers, urban area fraction, etc., for terrestrial ecosystems [25,40] or nutrient concentrations in aquatic ecosystems [41]. Other correlative models include generalized linear models, non-parametric multiplicative regression, decision trees, boosted regression trees, and artificial neural nets [34].

A further approach is that of the CLIMEX model (e.g., latest version Climex-Dymex 4.0.2), which was developed in the 1980s for predicting habitat suitability for invasive species. As described by Elith (2017) [34], this model is oriented around indices relevant to population growth (temperature, moisture, radiation, substrate, and behaviors like diapause), resistance to stresses (eight stressor indices based on temperature and moisture), and a few other indices related to being able to persist, such as the length of the growing season. The environmental data largely constitute weekly climatic conditions. This model can be regarded as largely mechanistic but also uses some particular information on a species that is not simply based on physiology.

Modelers of the invasive water hyacinth have used both modeling approaches. Modeling invasion potential at the global level, Cordeiro et al. (2009) [42] used MaxEnt version 3.3.3, with 19 bioclimatic variables. Kriticos and Brunei (2016) [11], using CLIMEX version 4, with a set of ten CLIMEX climatic variables, also addressed potential invasion at the global level, with an emphasis on water security. Several machine learning approaches (support vector, random forest, boosted regression tree) were used by Belayhun and Mekuriaw (2024) [43], with 10 vegetation and water indices, to predict the spread of water hyacinth in Lake Tana, Ethiopia. A classification tree model was used by Zarkami et al. (2002) [44] to predict water hyacinth spread in wetlands in Iran. Baker et al. used CLIMEX version 4 to develop a climatic suitability map to identify areas of high risk of water hyacinth invasion in Europe [45].

A weakness of the above modeling approaches, even those using CLIMEX, is that they cannot easily take into account possible interactions of the invasive species with native competitors [8] or natural enemies [7] that might limit its realized niche. Accounting for such biotic interactions is important, as doing so may limit range expansion. Interactions between native and invasive species can determine the success of an invasion. Native species often have already established competitive strategies and predator–prey relationships that can either hinder or facilitate the invasion process. For instance, if an invasive plant is highly palatable to native herbivores, it might spread more rapidly [46]. This is best determined through mechanistic modeling [47,48] at the level of population interactions. Mechanistic models of water hyacinth interactions with a biocontrol agent, the weevil *Neochetina crassipes*, exist [49], as well as a model of water hyacinth competing with a submersed aquatic macrophyte [50]. Here, we expand on these models to take into account several key environmental characteristics [51] that influence the ability of water hyacinth to successfully invade and, in particular, either exclude or coexist with native aquatic vegetation. We simulate the potential invasion of water hyacinth in a new habitat as a function of limiting nutrients, seasonally varying temperature [52], a competing submersed aquatic macrophyte, and aspects of spatial heterogeneity. We integrate and elaborate on components of existing models to achieve this.

In an earlier study, a spatially implicit model was developed to describe competition between a type of generic floating aquatic vegetation (FAV) and submersed aquatic vegetation (SAV) in a shallow waterbody [53] where the FAV was limited by a nutrient (generally nitrogen) in the water column but could limit solar radiation reaching the SAV. This model was parameterized for two aquatic vegetation species, floating duckweed (*Lemna gibba*) and submersed Nuttall’s waterweed (*Elodea nuttallii*) and tested experimentally. A subsequent extension of this model created to consider the effects of spatial extent and heterogeneity (spatially varying water depths) was developed by van Nes and Scheffer [54] using a spatially explicit lattice model with a 50 × 50 pixel grid. Van Nes and Scheffer’s model [53,54] showed that spatial heterogeneity in the form of varying water depths could extend the range of values of the limiting nutrient over which the FAV and SAV could coexist.

Temperature is also an important factor for the spread of aquatic plant species [55,56], including water hyacinth, which has a tropical origin. The effect of temperature, along with nitrogen, phosphorus, and other factors, on the growth rate of water hyacinth was studied by Wilson et al. (2005) [57] through data synthesis and modeling. Temperature is important in regard to the ability of water hyacinth to invade an aquatic system and compete, as temperature affects growth; in particular, frost has been shown to be a major cause of mortality of the leaves of water hyacinth. Therefore, in temperate areas, the competitive ability of water hyacinth fitness could be less than that of the native vegetation.

Xu and DeAngelis (2024) [50] used a spatially explicit model that is a variation of the spatial models created by van Nes and Scheffer (2005) [53,54] and McCann (2016) [58] to simulate competition between FAV and SAV, particularly competition between water hyacinth and Nuttall’s waterweed. The cited study was specific to Southern Florida, where the water temperature is favorable to water hyacinth. In this current study, we included the effects of temperature, both annual mean and seasonal variation, on the growth rate of water hyacinth using the equations for temperature effects developed by Wilson et al. (2005) [57] in the model. The purpose was to assess the risk of invasion of water hyacinth outside of its current range, specifically for areas with a lower mean annual temperature and larger seasonal variation, to mimic a possible spread northward. In addition to fluctuations in temperature, some aspects of spatial heterogeneity were built into the modeling. The overarching question is as follows: what conditions of the total limiting nutrients and annual temperature variation are suitable for the invasion of water hyacinth and, subsequently, its coexistence with native SAV (using Nuttall’s waterweed as a proxy in this case)?

## 2. Results

### 2.1. Effects of Temperature and Nutrients on Invasion and Coexistence

The model simulations showed three possible qualitative outcomes: a pure state for FAV, pure state for SAV, or a state of coexistence of the two species. Both the limiting nutrient *N* and temperature had an effect; high values of *N* and high values of mean annual temperature favored the FAV, while SAV was favored at the other ends of the spectrum of those values, and coexistence occurred for a range of intermediate values (the eight temperature scenarios of Table 1). The results for temperature scenario 3 (see Section 5), with a mean temperature of *θ_mean_* = 24 °C and a seasonal amplitude of *θ_amplitude_* = 7.5 °C, were close to the results obtained in [53]. In the cited authors’ spatially implicit model, coexistence for 24°C to 25 °C occurred between about nutrient *N* = 0.55 and *N* = 1.35, while for our temperature scenario 3, coexistence occurs across the discrete nutrient values from *N* = 0.7 to *N* = 1.2. In this scenario, the two species were approximately equal in terms of growth rates, a situation similar to that shown in Figure 1 (derived from [53]). As in [53], in our model scenario 3, for larger values of *N*, only FAV persisted, while for smaller *N* values, only SAV persisted. In Table 1, the mean temperature, *θ_mean_*, decreases and the temperature fluctuation amplitude, *θ_amplitude_*, increases as the temperature scenario number increases to the right in the table from a tropical to a more temperate climate.

The possibility of FAV excluding SAV disappeared by temperature scenario 4, with a mean temperature of *θ_mean_* = 22 °C and a seasonal amplitude of *θ_amplitude_* = 11 °C. However, coexistence was possible until scenario 8, with a mean temperature of *θ_mean_* = 16 °C and a seasonal amplitude of *θ_amplitude_* = 19 °C. This shows the significant effect of temperature on the growth rates of the two species and, consequently, on the competitive outcome.

### 2.2. Bifurcation Diagrams

Table 1 shows only the three qualitative states of competition between FAV and SAV. The quantitative values of biomasses are given in the bifurcation diagrams in Figure 1. The lines represent averages over both the whole spatial lattice as well as over the period of time after which the system reached a final state. Here, temperature scenarios 1 through 4 are shown. The mean temperature decreases and seasonal fluctuations increase when moving from panels (a) to (d). As the mean temperature decreases, the range of coexistence (where the red and blue lines overlap) shifts to the right (i.e., higher nutrient levels), and the final mean value of the FAV biomass substantially decreases. In these plots, the solid lines represent the biomasses of FAV (red) and SAV (blue) for simulations in which seasonal temperature fluctuations are included, while the dashed lines represent simulations where only the mean annual temperature was included.

### 2.3. Temporal Variability

The bifurcation diagrams show the biomasses averaged over both space and time. There is temporal variability of the biomasses, as shown in Figure 2 for two temperature scenarios in which coexistence occurred, namely, temperature scenarios 3 and 6. The cooler part of the season favored SAV, while the warmer favored FAV; thus, seasonal fluctuations occurred for the two species that were 180 degrees out of phase.

### 2.4. Spatial Heterogeneity

Spatial variability was included in the model, as vegetative propagules of FAV are randomly deposited in time in random spatial pixels in one of the scenarios. When conditions are favorable, the FAV propagules grow at the expense of SAV, as shown in Figure 3.

## 3. Discussion

Predictive models of potential invasions are needed to lay the groundwork for management actions [1]. The array of modeling approaches that have been applied to predicting invasions has been reviewed [1], and these approaches range from SDMs and statistical models [25,29] to population dynamics models [28] and cellular automata models [59]. As early as almost three decades ago, [60] developed a spatially explicit individual-based model for invasive plant spread, noting that such models can incorporate details of spatial heterogeneity and individual variability.

Our approach presented herein is spatially explicit, although it is not individual-based but follows biomasses in a 50 × 50 array of grid cells. The model allowed the incorporation of several key elements related to the question of whether an introduced species can successfully invade. It combined a climatic factor, temperature, with a factor that was more variable at a local scale, that is, the nutrient concentration in a particular waterbody. The model was oriented toward competition between an invader (FAV) and a native resident species (SAV), as competition with native species is one of the possible components of resistance to invasion that a native community can offer [61]. Because temporal variability is one of the environmental conditions that can affect population interactions [62], it was included through the simulation of seasonal variations in temperature (Figure 2). Finally, spatial heterogeneity and stochasticity were included by allowing propagules of the invasive species to be deposited at random times and places in the model grid that represents a shallow waterbody (Figure 3).

The results of the simulations showed that when the competitive abilities of the competing FAV and SAV were similar, a bifurcation diagram similar to that shown in Figure 2 (corresponding to the material from [53]) was the result; that is, the nutrient concentration determining which species was dominant or whether coexistence can occur depends on nutrient concentration in the same way. As the simulated temperature scenario became more temperate and thus further from the optimal temperature for the invader, the bifurcation diagram became distorted in favor of the native species, although coexistence was possible over almost the whole range of temperature scenarios if the nutrient level was high enough. However, coexistence virtually stopped in the temperature scenario corresponding to the seasonal temperature pattern of temperature scenario 8, close to the temperature patterns of Birmingham, Alabama. Although we did not make extensive comparisons with empirical data, it is important to note that the empirical data shown in Figure 2 (obtained by Kriticos and Brunel (2016) [11]) show the range of permanent occupations of water hyacinth in the Southeastern United States has an upper limit in Northern Alabama.

An important result shown in the bifurcation diagrams is that the seasonal temperature fluctuations can have a large effect on the mean biomasses of the examined species (Figure 1). This is especially true for the more temperate scenarios (Figure 1c,d), in which the seasonal fluctuations were relatively large. The fluctuations clearly promoted higher FAV biomass and lower SAV biomass, though they do not appear to affect the qualitative feature of whether invasion is possible.

The results of this study cannot in themselves be used to predict how far into regions with temperate climates water hyacinth might spread. This study is intended to simply predict the likelihood of water hyacinth invading a shallow lake in the presence of a particular submersed macrophyte that has a comparable growth rate but at a lower optimum temperature. Water hyacinth is known to occur transiently in areas of the Eastern United States with mean annual temperatures that are cooler than the temperature scenarios in our model. Our model only shows that temperatures lower than the optimum growth temperature for water hyacinth will slow its growth and decrease its competitive ability against better-adapted submersed vegetation. The range of nutrient levels, *N*, extended only to *N* = 2 mg L^−1^, so it did not include more eutrophic conditions. This study also suffers from the limitation that while the temperature dependance of the growth of the FAV, water hyacinth, is based on data, that of the native, more temperate species is simply assumed, though the other parameters for the species are similar to those of Nuttall’s waterweed. Therefore, the model study should be regarded as a theoretical exploration of how the combination of factors included in the model might affect water hyacinth’s ability to invade cooler habitats.

Our modeling approach consists of the use of a mechanistic population competition model and so is different from approaches employing the many correlation-type models or the CLIMEX model that have been in standard use over the last two decades. However, mechanistic approaches are frequently used. Higgins and Richardson (1996) [60] provided a review of the types of models of invasive organisms, including simple-demographic, spatial–phenomenological, or spatial–mechanistic models, including reaction diffusion models. Such mechanistic approaches will continue to be important in studying the details of the invasive process and how to control it. Spatially explicit individual-based models have been used to simulate interactions of the invasive *Melaleuca quinquenervia* tree in Southern Florida [63]. These models projected the outcome of the invader competing against five native species, slash pine, pond cypress, dahoon holly, sweet bay, and loblolly bay, along with the effects of a biocontrol agent on the invader.

The effects of temperature on the spread of water hyacinth in the Southeastern United States have also been studied using simpler models. For a site in Louisiana, Nesslage et al. (2016) [13] used a logistic model with a temperature-dependent seasonal mortality parameter and included a biological control in the model. This model projected a decline in the growth rate and survival of water hyacinth of 84% between 1976 and 2013 resulting from the combination of low winter temperatures and biocontrol measures. This shows that simple modeling approaches can be effective. Our model is more complex and thus requires more assumptions, which are rough estimates but are able to describe details involving the combination of temperature, nutrients, and competition, which may be useful for management.

## 4. Conclusions

We developed a spatially explicit model to explore the potential of an FAV, the water hyacinth, to spread to more-temperate areas. This model includes the effects of temperature and nutrients on the competition between water hyacinth and a submersed macrophyte. Although the ability of water hyacinth to invade decreases as the temperature scenario becomes less tropical and more temperate, for high levels of a given nutrient, invasion is possible over a wide range of temperatures. As the temperature pattern changed from tropical to cooler, more temperate levels, the competitive advantage shifted from the tropical FAV to the more temperate SAV, with a wide range across which coexistence occurred, which depended on the nutrient concentration. But even at the highest nutrient concentrations in the model, the FAV was unlikely to persist at current climates of latitudes in the Southeastern United State above that of Northern Alabama. This result may have some implications for where control efforts need to be concentrated. However, as the model used here was based on assumptions concerning the SAV that was competing with the FAV, the results should be seen as providing rough projections that should be tested empirically. Further work will involve surveying relevant empirical work for such tests.

## 5. Materials and Methods

Although the FAV in our model, *P. crassipes*, can generally be limited by either nitrogen or phosphorus, experiments conducted at the U.S. Department of Agriculture’s Invasive Plant Research Laboratory in Davie, Florida, show that *P. crassipes* growth responds strongly to nitrogen, which we assume is the limiting factor in the model. We follow Scheffer et al. (2003) [53] in assuming that SAV is light-limited. Figure 4 shows a schematic of the relationships. The inset represents the effect of temperature on the two species.

### 5.1. Model Developed by Scheffer et al. (2003)

Competition between two aquatic species, one SAV and one FAV, was modeled by [53] using the equations
(1)$$\frac{dS}{dt}=r_{s}*S*\frac{n}{n+h_{S}}*\frac{1}{1+a_{S}S+bF+W}-l_{S}*S,$$
(2)$$\frac{dF}{dt}=r_{F}*F*\frac{n}{n+h_{F}}*\frac{1}{1+a_{F}*F}-l_{F}*F,$$
where *S* and *F* are variables of the dry weight biomasses (g dW m^−2^) for SAV and FAV, respectively, and *n* is the nutrient concentration in the water column (mg L^−1^). The parameters *r_S_* and *r_F_* are the maximum growth rates, and *l_S_* and *l_F_* are the loss rates, including respiration and mortality, while *h_S_* and *h_F_* are the half-saturation values for the nutrient uptake of SAV and FAV. Parameters *a_S_* and *a_F_* are the intraspecific competitive effects of SAV and FAV, and *b* is the shading effect of FAV on SAV. Parameter *W* is the effect of the surrounding water’s light absorption on submersed plant growth.

Scheffer et al. (2003) [53] defined the total limiting nutrient concentration in the system as *N*, which includes this nutrient both in vegetation and as a solute in the water column. Therefore,
(3)$$N=n+nq_{S}S+{nq}_{F}F,$$
where *q_s_* and *q_f_* are coefficients of the effect of submerged and floating plants on the nitrogen concentration in the water column; that is, they represent the fractions of this nutrient tied up in the vegetation per unit dry weight biomass. The concentration, *n*, in the water, therefore, changes as vegetation biomasses changes:(4)$$n=\frac{N}{1+q_{s}S+q_{F}F}$$

In their model, [53] sought to examine the impact of varying *N* on the competition of SAV and FAV, so *N* was varied over a range of values. We used the equations of the model developed by [10], changing the growth of FAV from that of duckweed given in [53] (*r_F_* = 0.5) to *r_F_* = 0.1, which is consistent with many of the studies of water hyacinth surveyed in Wilson et al.’s work (2005) [57]. We changed the shading effect of the FAV on SAV from b = 0.01, which was appropriate for duckweed in [53], to *b* = 0.04, which reflects the higher density of *P. crassipes* vegetation. The SAV was again Nuttall’s waterweed.

### 5.2. McCann’s Cellular Automata (2016)

The model developed by [53] is spatially implicit. McCann (2016) [55] further developed this model into a spatial CA model composed of a grid of spatial pixels that could form two-dimensional areas of different shapes. We used a spatially explicit model, as it allowed us to initiate the development of two species in a variety of ways spatially, which could have made a difference in the outcomes yielded by the model. By dispersing propagules of each species in a random fashion across the spatial grid, the simulations provided a better chance of ensuring the competition did not depend too heavily on the starting biomasses, as might be the case in a spatially implicit model. It was assumed that SAV and FAV could spread among the pixels and that the limiting nutrient in the water column could diffuse. The edges of the spatial grid were reflective so that vegetation and the limiting nutrient did not spread out from the gridded system. We adapted the two-dimensional spatial model developed by [55] for the special case of a 50 × 50-pixel grid, with each pixel being 1 m × 1 m. This scale of resolution is the same as that employed by [55] and about the spatial scale of a few full-grown water hyacinth plants. Our model is not strictly a CA model, as continuous levels of biomasses of both species could occupy individual spatial cells.

### 5.3. Temperature Effects on Growth

The growth rates *r_F_* of the two species were assumed to depend on temperature, *θ*, in the form of a triangular function, *g_F_(θ)*, defined by a minimum threshold for growth, *θ_min,F_*, an optimal growth temperature, *θ_opt,F_*, and a maximum temperature, *θ_max,F_* [57].

### 5.4. Formatting of Mathematical Components


(5)
$$g_{F}\left(\theta \right)=\left\{\begin{array}{c} 0\;\;\;\;\;\;\;\;\;\;\;\;\;\;\;\;\;\;\;\;\theta \;\leq \;\theta_{min,F}\;\;\;\\ \frac{\theta -\theta_{min,F}}{\theta_{opt,F}-\theta_{min,F}}\;\theta_{min,F}<\theta <\theta_{opt,F}\\ \frac{\theta_{max.F}-\theta }{\theta_{max,F}-\theta_{opt,F}}\;\;\;\;\;\;\;\;\;\theta_{opt,F}<\theta . \end{array}\right.$$


The function *g(θ)* multiplies the *r_F_* of the water hyacinth, so that as *θ* varies, the effective growth rate varies with it. Temperate was simulated as a mean annual temperature and an amplitude of fluctuations:(6)$$\theta =\;\theta_{mean}+\theta_{amplitude}sin\left(\frac{2\pi t}{365}\right),$$
where the effects of a variety of values of *θ_mean_* and *θ_amplitude_* were used to test the ability of water hyacinth to invade and possibly exclude the native species. The effective growth rate of the FAV, water hyacinth, was thus
(7)$$r_{F,effective}=g_{F}\left(\theta \right)r_{F}.$$

We made a similar assumption of a triangular temperature dependence, *g_S_(θ)*, on the SAV. We did not know the temperature dependence of Nuttall’s waterweed, but we assumed there was an influence of a set of parameters *θ_min,S_*, *θ_opt,S_*, and *θ_max,S_* that describe a more sub-tropical to temperate species, such that the triangular function is similar in shape to that of the water hyacinth but shifted towards lower temperatures (Figure 5).

### 5.5. Seasonal Temperatures

The purpose of this study was to estimate the risk of water hyacinth invading regions with climates outside its tropical to sub-tropical range. Therefore, we simulated a series of climates ranging from sub-tropical, similar to Miami, Florida, to warm temperate, similar to Birmingham, Alabama (Figure 6). The climates differ both in terms of mean annual temperature and the amplitudes of seasonal temperatures, which are larger in the more temperate regions, especially those with cooler winter temperatures.

### 5.6. Parameterization of the Model

The parameters of our model generally correspond to those used by Scheffer et al. (2003) [53] and McCann (2016) [55], though the growth rate was modeled for water hyacinth—including regarding temperature effects obtained using data from Wilson et al.’s work (2005) [57]—rather than duckweed (*Lemna gibba*) as modeled in those papers to determine the spatial dynamics of FAV (duckweed) and SAV (Nuttall’s waterweed). The parameter values are listed in Table 3.

### 5.7. Simulations

Simulations were performed to approximate the way that invasion might occur. As both species can propagate vegetatively, small vegetative propagules, 0.1 g dW of each species, were scattered randomly both temporally and spatially. The frequency of the invasive FAV was low, slightly less than two vegetative propagules per year over the whole lattice. Propagules of the native species were input randomly at a much higher frequency.

Twenty values of *N* were used in the simulations, ranging from 0.1 to 2.0 mg L^−1^, identical to the range of nutrient values used by Scheffer et al. (2003) [53], along with eight seasonal temperature scenarios (shown in Figure 6). The annual means and associated seasonal fluctuations of the eight temperature scenarios are arranged in order from tropical to warm temperatures in Table 2. Simulations were run for a period corresponding to over twenty years in order for the system to reach a final state.

## Figures and Tables

**Figure 1 plants-13-02621-f001:**
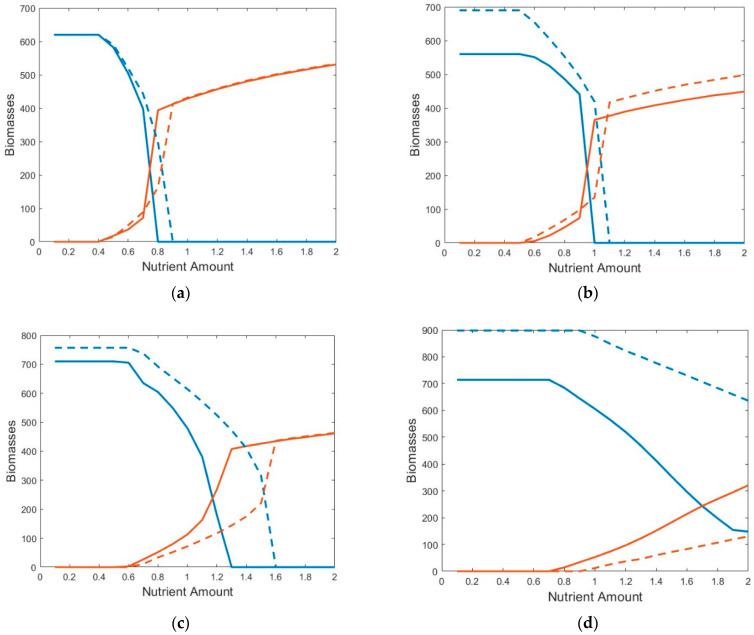
Bifurcation diagrams showing the biomasses (g dW m^−2^) of FAV (red lines) and SAV (blue lines) as a function of total nutrient concentration (mg L^−1^). (**a**) temperature scenario 1 (*θ_mean_* = 27 °C with a seasonal amplitude of *θ_amplitude_* = 4 °C), (**b**) temperature scenario 2 (*θ_mean_* = 25.75 °C with a seasonal amplitude of *θ_amplitude_* = 6 °C), (**c**) temperature scenario 3 (*θ_mean_* = 24 °C with a seasonal amplitude of *θ_amplitude_* = 7.5 °C), and (**d**) temperature scenario 4 (*θ_mean_* = 22 °C with a seasonal amplitude of *θ_amplitude_* = 11 °C). The dashed lines represent biomasses at only the mean annual temperatures, without fluctuations.

**Figure 2 plants-13-02621-f002:**
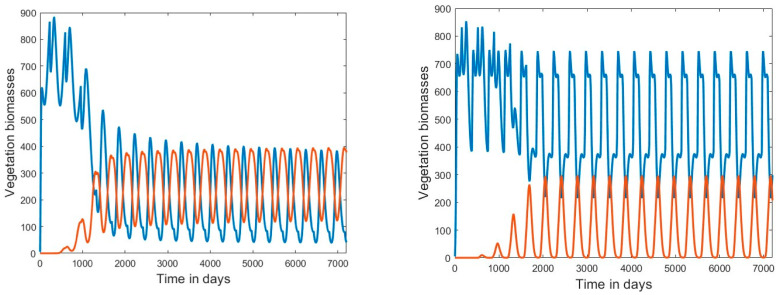
Variation in biomasses (g dW m^−2^) of SAV (blue lines) and FAV (red lines) over time. (**Left**) Temperature scenario 3 (*θ_mean_* = 24 °C with seasonal amplitude of *θ_amplitude_* = 7.5 °C), with *N* = 1.2; (**Right**) temperature scenario 6 (*θ_mean_* = 18.5 °C with seasonal amplitude of *θ_amplitude_* = 16 °C), with *N* = 1.

**Figure 3 plants-13-02621-f003:**
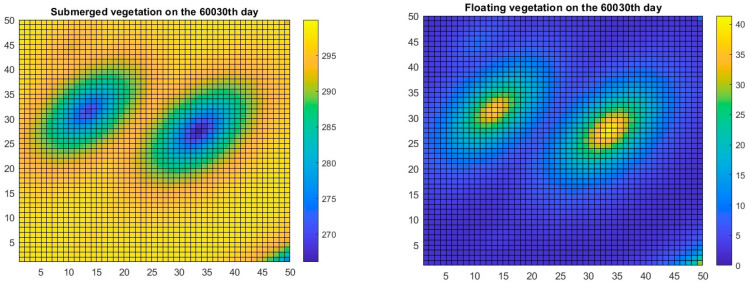
The spatial distribution of vegetation biomasses is shown for SAV (**left**) and FAV (**right**) sometime after two propagules of FAV had been deposited in random pixels and started to spread.

**Figure 4 plants-13-02621-f004:**
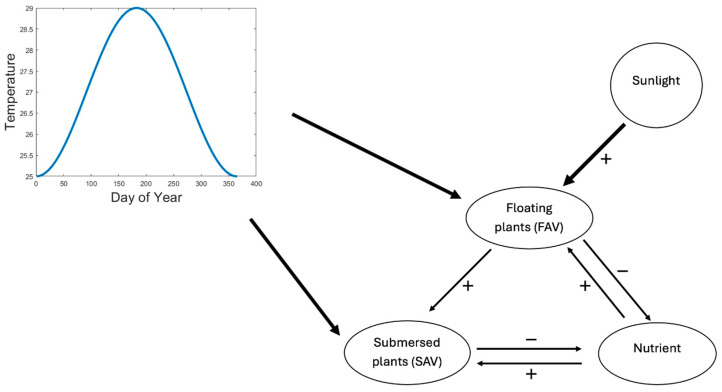
Competition between FAV and SAV. The SAV is a better competitor for the limiting nutrient (nitrogen), but the FAV can shade the submerged plant. The inset represents the effect of temperature on both plant species.

**Figure 5 plants-13-02621-f005:**
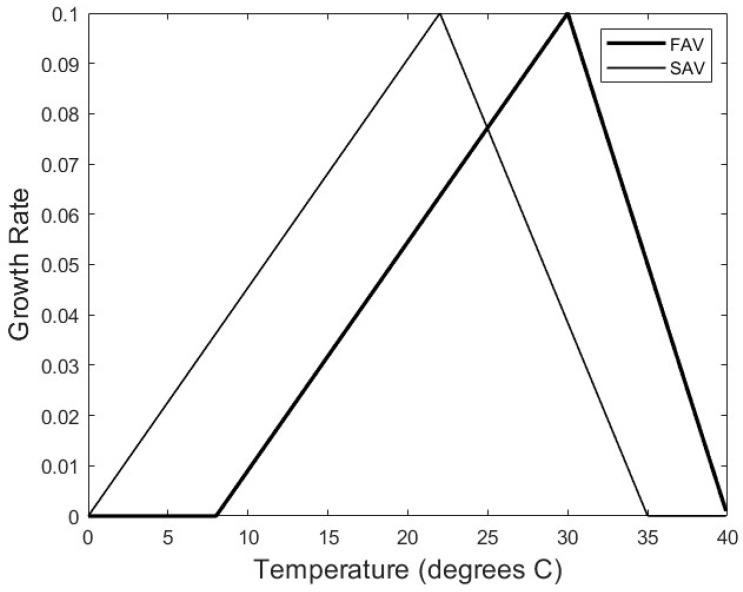
Growth rate vs. temperature for floating (FAV) and submersed (SAV) macrophytes. FAV triangle is based on Figure 5 of Wilson et al. (2005) [57], while SAV is assumed to correspond to a temperate species similar to Nuttall’s waterweed with a lower temperature growth optimum.

**Figure 6 plants-13-02621-f006:**
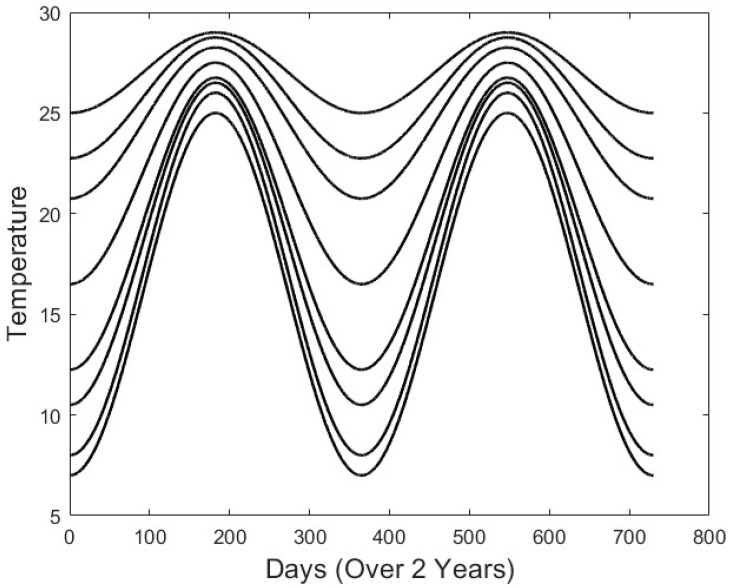
Eight temperature scenarios used in the simulations. The mean temperatures and amplitudes are given in Table 2. The extreme upper and lower curves were estimated based on graphs based on a Google search of ‘Climate in Miami, Florida’ and ‘Climate of Birmingham, Alabama’. The curves were fit based on averages of monthly highs and lows. The curves between the extremes were found to exhibit an arbitrary series of values of means and the amplitudes of the temperatures between those.

**Table 1 plants-13-02621-t001:** Results regarding the invasion potential of FAV and possible coexistence with SAV.

Nutrient *N* ^†^	Temperature Scenarios (Mean Temperatures, Celsius)							
	27°	25.75°	24°	22°	19.5°	18.5°	17°	16°
2.0	F	F	F	C	C	C	C	C
1.9	F	F	F	C	C	C	C	S
1.8	F	F	F	C	C	C	C	S
1.7	F	F	F	C	C	C	C	S
1.6	F	F	F	C	C	C	C	S
1.5	F	F	F	C	C	C	C	S
1.4	F	F	F	C	C	C	C	S
1.3	F	F	F	C	C	C	C	S
1.2	F	F	C	C	C	C	S	S
1.1	F	F	C	C	C	C	S	S
1.0	F	F	C	C	C	S	S	S
0.9	F	C	C	C	S	S	S	S
0.8	F	C	C	C	S	S	S	S
0.7	C	C	C	S	S	S	S	S
0.6	C	C	S	S	S	S	S	S
0.5	C	S	S	S	S	S	S	S
0.4	S	S	S	S	S	S	S	S
0.3	S	S	S	S	S	S	S	S
0.2	S	S	S	S	S	S	S	S
0.1	S	S	S	S	S	S	S	S

F = only invader FAV persists; S = only native SAV persists; C = coexistence. ^†^ mg L^−1^.

**Table 2 plants-13-02621-t002:** Values of parameters changed in the simulations. *θ_mean_* and *θ_amplitude_* values are ordered from tropical to warm temperate climates.

Parameter	Definition	Values	Units
*N*	Total nutrient concentration (nitrogen) in the system	0.1, 0.2, …, 2.0	mg L^−1^
*θ_mean_*	Mean annual temperatures	27.0, 25.75, 24.5, 22.0, 19.5, 18.5, 17.0, 16.0	deg C
*θ_amplitude_*	Amplitudes of seasonal fluctuations	4, 6, 7.5, 11, 14.5, 16, 18, 19	deg C

**Table 3 plants-13-02621-t003:** Parameter values for reproduction, seedling dispersal, and litter suppression of seedlings.

Parameter	Definition	Value	Units
*r*	Maximum growth rates of SAV and FAV	0.1, 0.1	day^−1^
*h*	Half-saturation of *n* for SAV and FAV	0.0, 0.2	mg L^−1^
*a*	Self (light)-limitation	0.1	1/g dW m^−2^
*l*	Losses to SAV and FAV	0.05, 0.01	day^−1^
*b*	Shading effect of FAV on SAV	0.04	1/g dW m^−2^
*W*	Light attenuation in the water column	0	unitless
*θ_min,F_*, *θ_min,S_*	Minimum temperature for growth	8, 0	deg C
*θ_opt,F_*, *θ_opt,S_*	Optimum temperature for growth	30, 22	deg C
*θ_max,F_*, *θ_max,S_*	Maximum temperature for growth	40, 35	deg C
*N*	Total nitrogen in system	Various	mg L^−1^
*q_s_*	Nutrient per unit biomass SAV	0.025	mg/g dW
*q_f_*	Nutrient per unit biomass FAV	0.005	mg/g dW

## Data Availability

The study does not use any data.

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
