# Peer review of "Modeling the Effects of Temperature and Limiting Nutrients on the Competition of an Invasive Floating Plant, Pontederia crassipes, with Submersed Vegetation in a Shallow Lake"

_plants, 2024, doi:10.3390/plants13182621_

Round 1

Reviewer 1 Report

Comments and Suggestions for Authors

Dear Editor The manuscript submitted for review addresses the important issue of the spread of invasive plants in water bodies. Modeling such a phenomenon is essential in assessing changes in biodiversity in aquatic ecosystems and water quality. The authors of the text took into account the temperature factor and its changes, as well as various concentrations of nitrogen in the water column. The analyses included a very low concentration of this element, which is both an advantage and a disadvantage of these considerations. Please expand on this issue in the introduction (low concentration of N, nitrate form or ammonium form?) based on your own research or research presented in other scientific articles. I also lack justification for eliminating the second very important nutrient, phosphorus, from the considerations. Please also comment on the importance of other macro- and micronutrients on the growth of aquatic plants.

Best regards

Reviewer

Reviewer 2 Report

Comments and Suggestions for Authors

Review for the paper “Modeling effects of temperature and limiting nutrient on competition of an invasive floating plant, Pontederia crassipes, with submersed vegetation in a shallow lake” by Linhao Xu and Donald L. DeAngelis submitted to “Plants”.

The authors of this research paper conducted an analysis to assess the risk of expansion of the water hyacinth Pontederia crassipes, an invasive aquatic plant species, beyond its present range in the United States. They developed a spatially explicit model that simulated the invasion of water hyacinth into habitats with different mean annual temperatures from its own growth optimum, as well as seasonal fluctuations in temperature. The model also considered 12 different nutrient concentrations and eight 8 temperature scenarios with varying mean annual amplitudes of seasonal temperature variation around the mean of the invaded habitat. They found that the results exhibited large differences from theoretical expectations based on competition purely for light and nutrients. The model was able to determine the ability of water hyacinth to invade and either exclude or coexist with the native vegetation under the various environmental conditions simulated.

The results of this study may have important implications for understanding the factors that influence the potential for a non-native plant species to invade a new habitat. By considering both broad-scale factors, such as climate, and local factors, such as nutrient availability and biotic community, the model provides a more comprehensive assessment of invasion risk. This information could be valuable in developing strategies to prevent or manage the spread of invasive species such as water hyacinth.

The study highlights the importance of considering the complex interactions between environmental factors and the competitive dynamics between invasive and native species. Overall, this research contributes to our understanding of the factors that influence the invasion potential of non-native plant species and provides a framework for assessing invasion risk in different environmental contexts. The knowledge gained from this study could inform conservation and management efforts aimed at protecting ecosystems from the detrimental effects of invasive species.

Despite the importance of this topic, the authors did not include important background information and a detailed description of the methodology used. The discussion is brief and needs further revision. Many statements need additional citations.

Recommendations.

Abstract.

The authors provided little information on their main results and outcomes. They should updated this section with relevant data, for example two scenarios including one most dangerous and the other more optimal for the recipient ecosystem.

The authors should provide an explicit answer to their question of what conditions of total limiting nutrient and annual temperature variation are suitable for invasion of water hyacinth, and then, of coexistence with native SAV as highlighted in L 105-108.

In the abstract, the authors should also indicate the significance of their results for monitoring of invasive plant populations and conservation of local populations.

Introduction.

The text briefly touches on the differences between species distribution models and mechanistic models but does not elaborate on how each method works in detail. The authors should provide examples of each model's methodology.

L 34-37. The authors should include supporting citations for this text.

L 35. The text refers to "coarse scale indices" but does not specify what these include beyond climate conditions. Clarifying the specific variables involved in these models would enhance the reader's understanding.

L 38-44. The authors should provide some examples illustrating these statements with corresponding citations.

L 39. While the text mentions the importance of considering interactions with native species, it would be helpful to include examples or case studies where these interactions have significantly impacted the invasion process.

L 45-53. The authors should include supporting citations for this text.

L 48. Why January and June? Many geographic areas have different extremes.

L 64-69. The authors should include supporting citations for this text.

Methods.

L 227-229. The authors should justify these assumptions with citations.

It is unclear what the left panel means, as it is provided without any comments and citations.

L 251. The authors should specify these "minor changes" in parameters.

L 255-262. The authors should clarify the differences in results and advantages or disadvantages of using a spatial CA model compared to a spatially implicit one.

L 261. The authors should explain the choice of grid size and indicate whether the results are sensitive to changes in grid size or shape.

L 267. Please, check the title number.

L 278 and Figure 5. What sources were used to construct these growth patterns? How were the parameter values for Nuttall's waterweed determined in the absence of specific temperature dependence data? A more detailed justification for these assumptions and any literature support are necessary.

Section 5.4 and figure 6. What sources were used to construct this figure?

L 315. A footer is needed.

Results.

Table 1. The authors should include temperatures instead of numbers of scenarios for better understanding.

Figures 1 and 2. The authors should provide units for biomasses.

Figure 2 caption, The authors should explain the meaning of red and blue lines.

Discussion.

This section should be supplemented by more detailed comparisons with similar studies, highlighting the novelty of the authors' data. The authors should highlight the implications of their results for management and conservation.

In the "Conclusion", the authors should 1) provide more details on the model predictions, such as the specific temperature and nutrient ranges that favor water hyacinth invasion and how seasonality affects its biomass, 2) include a statement on how the model results could inform management strategies to control or mitigate water hyacinth invasions, especially in temperate regions, and 3) mention limitations of the model and 4) indicate further research on this topic.

Round 2

Reviewer 2 Report

Comments and Suggestions for Authors

The authors have revised the paper according to my comments.

Accept.